

# Designing wind turbines for profitability in the day-ahead markets

Mihir Mehta, Michiel Zaaijer, and Dominic von Terzi

Wind Energy Group, Faculty of Aerospace Engineering, Delft University of Technology, Netherlands

**Correspondence:** Mihir Mehta (m.k.mehta@tudelft.nl)

**Abstract.** Traditionally, wind turbine and wind farm designs have been optimized to minimize the cost of energy. Such a design would make sense when bidding in price-based auctions. However, in a future with a high share of renewables and zero subsidies, the wind farm developer is exposed to the volatility of market prices, where the price paid per kWh of energy would not be constant anymore. The developer might then have to maximize the revenue earned by participating in different

energy, capacity, or ancillary services markets. In such a scenario, a turbine designed for maximizing its market value could be more profitable for the developer compared to a turbine designed for minimizing the Levelized Cost of Electricity (LCoE). This study is in line with this paradigm shift in the field of turbine and farm design. It is a continuation of a previous study conducted by the same authors (Mehta et al., 2023), which explicitly focused on the drivers for turbine sizing w.r.t. LCoE. The goal of this study is to optimize the design for a new set of objective functions and analyze how various day-ahead market

conditions and objectives drive turbine design. A simplified market model that can generate hourly day-ahead market prices is developed and coupled with a wind farm-level Multi-disciplinary Design Analysis and Optimization (MDAO) framework to evaluate key economic indicators of the wind farm. The results show how the optimum turbine design is driven by both the choice of the economic metric and the market scenario. However, an LCoE-optimized design is found to perform well w.r.t. profitability-based economic metrics like MIRR/PI, indicating a limited need to redesign turbines for a specific day-ahead

market scenario.

**Keywords:** Offshore wind, Wind turbine design for profits, Wind farm design, energy markets, Multi-disciplinary design optimization and analysis (MDAO), beyond Levelized Cost of Electricity (LCoE).

## 1  Introduction

The share of renewables has now reached almost 30% of the total electricity generation, wind energy being the fastest-growing

technology (International Energy Agency, 2021). This share is expected to grow even faster with the rapidly falling costs and better designs. For some announced tenders in the North Sea, the Levelized Cost of Electricity (LCoE) is already in the range of 50-60 €/MWh (Wind & water works, 2022; Lensink and Pisca, 2019). The fall in the costs so far has been due to the reductions in the Operations & Maintenance costs and the continuous upscaling of turbines (Lantz et al., 2012; International Renewable Energy Agency (IRENA), 2019; Veers et al., 2019). However, the system dynamics and market incentives are also

rapidly changing resulting in newer design objectives and constraints. This may demand a change in the turbine and farm design philosophy.





At first, turbines were new elements in the grid system and needed to be demonstrated and developed on many fronts. They were valued for their score on primary performance indicators, such as reliability and the Annual Energy Production (AEP). One of the consequences of this was a focus on the aerodynamic performance of the turbine, executed via maximizing the

power coefficient ($c_P$) of the rotor (Chehouri et al., 2015). However, this metric would ignore the mass (and costs) of the rotor involved resulting in relatively heavy structures. Then, turbines and farms were commercialized, but with support schemes that effectively resulted in an (almost) constant value of produced electricity. This led to a focus on the minimization of LCoE. Also, LCoE is a metric that is easy to calculate, covers all the aspects of a wind farm, and is hence universal in nature. Various wind farms across different sites or even different technologies could be compared simply by looking at the LCoE values. Also, in

subsidy-based auctions or Power Purchase Agreements (PPA) where nearly a fixed electricity price is ensured, minimizing the LCoE would effectively correspond with maximizing profit. In this era, turbines were often optimized for the support scheme, such as yielding exactly the amount of full-load hours that were subsidized in a year.

In a subsidy-free environment, the developer is exposed to the volatility of the market prices. This goes away from the traditional subsidy-based approach where the wind farm developer would be ensured a fixed premium or price. Due to the

merit-order effect in the day-ahead market, regions with a high wind penetration, quite often, displace the expensive generators during times of high winds, resulting in low prices. This effect is also known as the cannibalization effect. The drop in the market value of wind with an increasing share of renewables has been shown in several studies such as Mills and Wiser (2012) and Hirth (2013). As market prices negatively correlate with grid-wide average wind speed (cannibalization), turbines should not only be designed to reduce costs but also to increase the value of the produced electricity. Shields et al. (2021) performed

an extensive study showing the benefits of upscaling turbines and farms to reduce the LCoE. Some of the shortcomings of the study are addressed in Mehta et al. (2023). However, both studies are focused on the LCoE and do not include market prices.

Since LCoE, as a metric, does not capture the varying electricity price per kWh, the market value of wind goes unaccounted for, and this is why there is a need to look beyond LCoE (Loth et al., 2022). This has led to the expectation that such market-driven designs have larger rotors, to generate more electricity at high prices, during low-wind-speed periods. Some studies

propose very low specific power turbines that produce high power at lower wind speeds and also cut-out earlier when conventional wind turbines reach their rated power. This results in higher revenues in return, and are also beneficial to the electricity system as they result in better system adequacy (Hirth and Müller, 2016; Swisher et al., 2022). Chen and Thiringer (2017) include market prices and look at leveraging overplanting and curtailment to increase wind farm profits. However, the study looks at absolute profits and does not look at all the changing cost elements in a wind farm. However, to comment on the

profitability of a given turbine design, a comprehensive analysis taking into account all the cost benefits and revenue gains at a wind farm level is required. There is little consensus on whether the discussed concepts reap higher economic benefits (using profitability metrics beyond LCoE) for a wind farm developer.

Profitability metrics like Internal Rate of Return (IRR), Net Present Value (NPV), Profitability Index (PI), etc. that include both costs and revenues are commonly used to assess the economic performance of wind farm projects (de Oliveira et al., 2011).

Some other metrics like Value Factor (VF) and the Cost of Valued Energy (CoVE), formulated by Simpson et al. (2020), also take into account the market value of wind. Since each metric has a different formulation, the economic performance of the





wind farm depends on the choice of the economic metric, which poses an additional challenge w.r.t. wind turbine design optimization. This study tries to address these gaps by exploring how turbines should be sized for subsidy-free markets. The term 'markets' refers to different possible future realizations of the day-ahead market, where the bulk of the electricity is traded.

The research question can hence be formulated as:

*How do wind turbine size and specific power change, w.r.t. an LCoE-optimized turbine, when maximizing its economic value in the day-ahead market?*

To answer the main question, two sub-questions are formulated that will be addressed in this work.

1. How do various economic metrics, that include the market value of wind energy, drive turbine design?

2. How do different day-ahead market price scenarios drive turbine design?

The turbine size and specific power refers to two main system-level parameters of a turbine, the rated power and rotor diameter. These are the two design variables that are optimized in this study. It should be noted that this study looks at the
wind farm developer's perspective and only includes revenues from the day-ahead (spot) market, excluding revenues from any capacity payments and grid (or other ancillary) services. The share of revenue may shift from energy markets to capacity or ancillary services markets in a future with high penetrations of renewables (Dykes, 2020). We already see subsidy-free offshore wind farms coming up that will be exposed to variable market prices (Rijksdienst voor Ondernemend Nederland, 2020). With this paradigm shift, it then becomes crucial to revisit the design philosophy used for turbine and farm optimization.

In the following chapters, a generic modeling approach is explained, followed by the optimization problem formulation and the results for various market scenarios and economic metrics (as objective functions). This research is a follow-up of Mehta et al. (2023), which explicitly focused on turbine sizing from an LCoE perspective. Hence, the models w.r.t. the wind farm elements are used as previously developed and are complemented with a market model to simulate the revenue-based objectives.

## 2 Modelling approach

This section discusses the general approach used in this study to model the various elements of the wind farm and the day-ahead markets. A Multi-disciplinary Design Analysis and Optimization (MDAO) based approach is used where all the disciplines of a wind farm are coupled. As a result, the trade-offs occurring at a farm level are captured. Ashuri et al. (2016), Perez-Moreno et al. (2018), Dykes et al. (2018), Bortolotti et al. (2022) and many other studies have explored the benefits of MDAO in the
wind energy domain, either at a turbine or at a farm level.

For this study, an eXtended Design Structure Matrix (XDSM) of the framework used to evaluate all the wind-farm level parameters is shown in Fig. 1. It was initially developed by Sanchez Perez Moreno (2019) and later on modified by several researchers. In this study, the turbine size, represented by the rated power and the rotor diameter, is optimized. To find the





optimal turbine size, the framework is run as an analysis block. For each set of design variables and a given market scenario,
the framework is executed and the economic performance of the wind farm is evaluated. The design variables along with
some user-defined inputs are fed into the models to evaluate the hourly farm power and the costs of various farm elements. A
simplified market model is developed and added to the framework to quickly generate hourly spot prices for a specified day-
ahead market scenario. The costs and farm power from the wind farm framework along with the spot prices from the market
model give the cashflows of the project and hence, several economic indicators like the Internal Rate of Return (IRR), Net
Present Value (NPV), etc., can be evaluated. The framework has all the elements needed to evaluate the LCoE of a wind farm.
The turbine design optimized for LCoE serves as a baseline against which, the designs optimized for various market scenarios
can be compared.

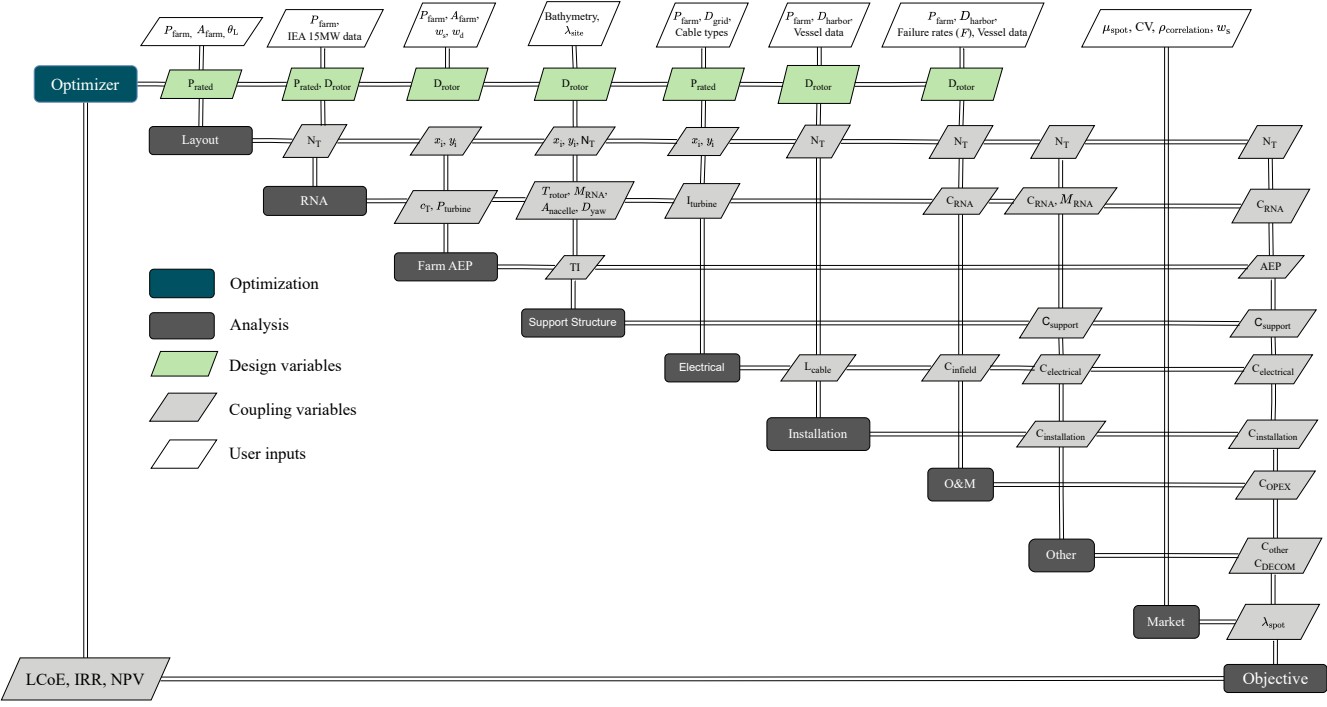

**Figure 1.** XDSM of the wind-farm level MDAO framework

## 2.1   Wind farm elements

As mentioned in the introduction, the modeling framework w.r.t. the wind farm elements (CAPEX of different components,
wake loss estimation, farm OPEX, etc.) is adopted from Mehta et al. (2023). That research provides an elaborate description
of the models and their implementation. Hence, in this research, the wind farm aspects that are included in the framework are
only briefly summarized in this section.





A change in the design variables (rated power and rotor diameter of the turbine) leads to significant changes across the wind farm. The framework's purpose is to capture the dependencies of each discipline on the design variables, while also capturing the interactions between different disciplines. The IEA 15 MW turbine (Gaertner et al., 2020) is used as the reference for all turbine-related costs. Inputs for the vessel data and failure rates, required for the installation and operations and maintenance (O&M) costs of the farm, are based on Dinwoodie et al. (2015), Smart et al. (2016), BVG Associates (2019), Shields et al. (2021), and Mangat et al. (2022). The functionalities of the models are explained below.

- **Rotor Nacelle Assembly (RNA):** The RNA module determines the performance curves for the turbine along with the cost of several components like the rotor, hub, generator, etc. The aerodynamic and structural properties are scaled from the IEA 15MW reference turbine. Most of the RNA costs are scaled w.r.t. the mass, which is either scaled from the reference turbine or, for some components, derived using the DrivetrainSE model (NREL, 2015). The rotor mass is scaled with the rotor diameter and adjusted for the changes in thrust w.r.t. the reference turbine, while the generator mass is scaled with the turbine's rated torque.

- **Layout:** The layout module generates a regular square layout based on the area constraint, the number of turbines in the farm, and the orientation of the layout (determined by the dominant wind direction).

- **Farm AEP:** The wake losses are calculated using the Bastankhah Gaussian model (Bastankhah and Porté-Agel, 2014) from the PyWake library of Pedersen et al. (2019). The thrust curve of the turbines, normalized spacing between the turbines, and the wind conditions determine the wind speed deficit at each turbine. The power curve of the turbine is then used to determine the turbine power production, which is eventually summed up for all the turbines to give the overall farm AEP.

- **Support structure:** The support structure module, used to determine the costs of the tower and monopiles, is based on the work of Zaaijer (2013). The hub height is determined by the rotor radius and the clearance of the blade from the water. The aerodynamic and hydrodynamic loads are used to determine the dimensions of the tower and the foundation.

- **Electrical:** The electrical module determines the cost of the array cables, the substation, and the export cable. The cost of the infield cables is driven by the farm layout and the turbine rated current, while the cost of the substation and export cable is driven by the total farm power.

- **Installation:** This module calculates the costs for installing the turbines, foundations, and the electrical system. The assumptions around the foundation installation time, turbine installation time, transit time, cable laying and burial rate, etc. determine the total days of operation for the respective vessels. The total installation time and the vessel day rates result in the total installation costs.

- **Operations & Maintenance (O&M):** The O&M cost module determines the annual costs for both preventive and corrective maintenance of the farm, while also assuming some fixed operational costs. The number of vessel trips and





spare part costs depend on the failure rates, the number of turbines in the farm, and the type of maintenance. The total
O&M costs are obtained by summing up the operational, vessel, spare part, and technician costs.

– **Other costs:** The other wind farm costs include project development costs, other turbine costs, contingency, decommissioning, etc. These are assumed to have a fixed percentage share in the total farm CAPEX. The decommissioning costs, however, are based on the number of turbines to be removed, the RNA mass, hub height, and the cable length.

The framework is completely open-source (Mehta, 2023). Further details about the models in the framework and their implementation are discussed in Mehta et al. (2023).

## 2.2 Market model

The hourly prices for spot markets can be simulated using complex market models like Balmorel (Wiese et al., 2018) or EMMA (Hirth et al., 2021). These are energy system models that minimize the system cost required to satisfy the demand. To simulate a future scenario, various inputs like electricity demand, capacity and costs of various generation technologies, fuel costs, cross-border trade, carbon prices, etc. are required. This enables the model to capture the complex market effects. However, it also makes it difficult to use such models to quickly simulate hundreds of future price scenarios to evaluate a business case of a project. Verstraten and van der Weijde (2023) argue that these complex models can be used as benchmarks while simpler models can be used to assess renewable business cases. The authors show the effect of change in the capacity of various technologies on the market clearing price using an in-house stochastic market simulator. However, the tool still requires information about the capacities of different assets, their operational strategies, and the electricity demand as inputs. For this study, it is important to capture the cannibalization effect of wind. From a turbine design perspective for a given wind farm, this translates to the relation between spot market prices and wind speeds. The purpose of the market model is not to accurately predict spot prices for a given year in the future. Instead, the purpose is to have a parameterized model to generate spot prices where the model parameters can be easily varied to simulate various future market scenarios. The generated spot price data can then be used to determine annual revenues.

The relation between spot prices and wind speed can be represented with the help of a univariate model that uses a linear or a polynomial fit. However, it is difficult to comment on how the coefficients would evolve in the future. Hence, this study uses a different approach to model the spot prices. Fig. 2 shows the distribution of the spot prices for the years 2016-2020 for both Denmark (left) and the Netherlands (right), taken from the European Network of Transmission System Operators for Electricity (ENTSO-E) (2023). While the Netherlands observed a higher mean value of spot prices than Denmark (due to a relatively lower renewable penetration), the standard deviation was the same. It can be seen that the spot price distribution can be approximated by a normal distribution. This approximation is later verified w.r.t. how it affects the annual revenues of a wind farm. Instead of using the absolute standard deviation, the spread around the mean can also be expressed in the form of a coefficient of variation (CV), i.e. the ratio of the standard deviation to the mean. Thus, with a constant CV, the variability in the price increases/decreases along with the mean price.



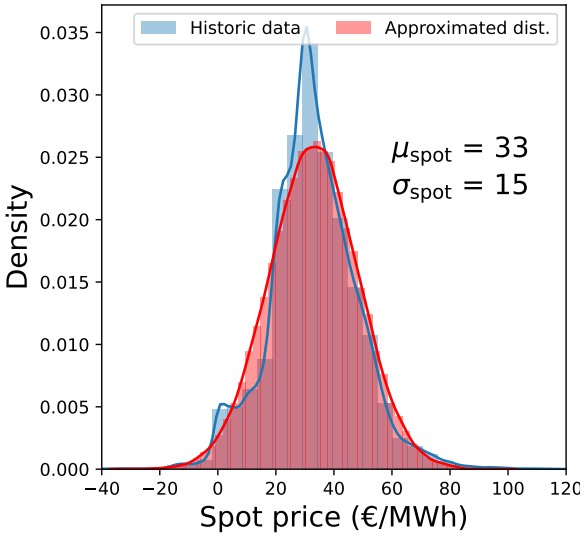
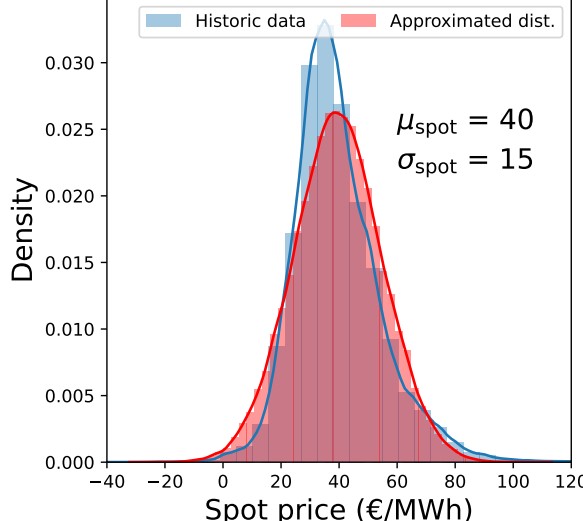

**Figure 2.** Spot price distribution and its approximation to a normal distribution for DK (left) and NL (right)

Also, due to the cannibalization effect, spot prices have a negative correlation with wind generation. An illustration of the cannibalization effect, for Denmark (left) and the Netherlands (right), is shown in Fig. 3, where the spot prices for the years 2016-2019 are plotted against the wind resource for a given site. As the wind energy penetration in Denmark is higher, compared to that of the Netherlands, it experiences a higher negative correlation between spot prices and wind speeds. The spot price data ($\lambda_{\mathrm{spot}}$) to be generated can be expressed as a function of various parameters shown in Eq. (1), where $\mu_{\mathrm{spot}}$ is the mean of the normal distribution for spot prices, CV is the coefficient of variation, and $\rho_{\mathrm{correlation}}$ is the correlation coefficient between spot prices and the site-specific wind speeds ($w_{\mathrm{s}}$).

$$\lambda_{\mathrm{spot}} = f(\mu_{\mathrm{spot}}, \mathrm{CV}, \rho_{\mathrm{correlation}}, w_{\mathrm{s}}) \tag{1}$$

The correlation coefficient only has an effect when the standard deviation is high enough. For low standard deviations, the correlation coefficient has no meaning. A low value of CV, which corresponds to a lower standard deviation, results in a smaller spread of data around the mean. As a consequence, for values of CV close to 0, even a high negative correlation of -1 would result in no variations of the spot prices w.r.t. the wind speed. This effect is shown in Fig. 4 where the generated spot price data for a given mean and a high negative correlation is plotted, for two different values of CV. It can be seen that for low values of CV, the spot prices do not change much. This effect is also shown for time series data of a week where the spot prices for a lower CV (in orange) don't change much even for large fluctuations in the wind speed (in black).

In this study, the CV is kept constant. It is known that the value for CV also differs, but it is expected to have the smallest range of variability of all the three market parameters. Keeping it constant simplifies the model, while still being able to capture the most relevant variations. The spot prices can be generated by sampling data from the normal distribution (defined



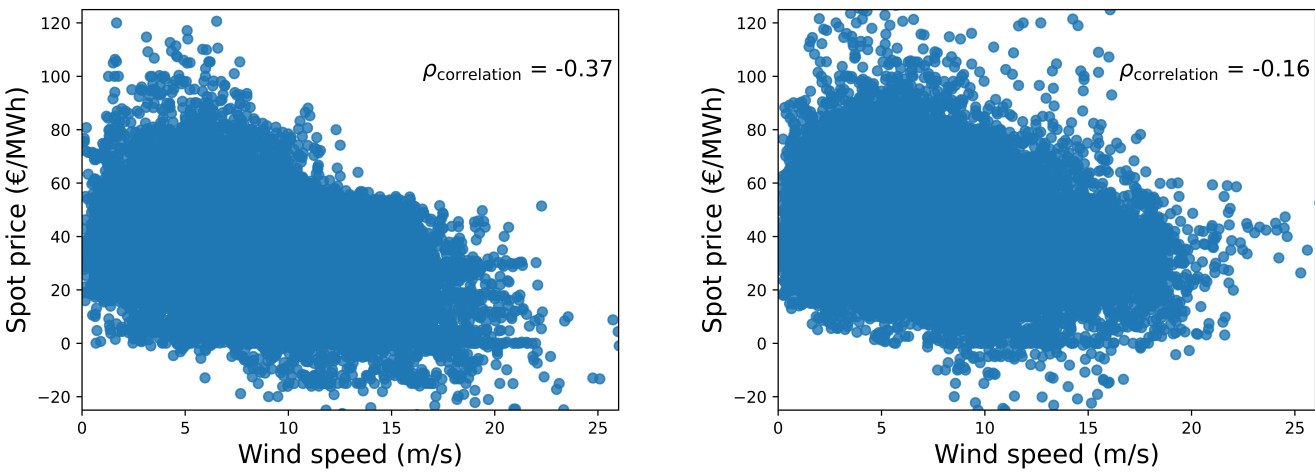

**Figure 3.** Correlation between spot prices and site-specific wind speeds for Denmark (left) and the Netherlands (right)

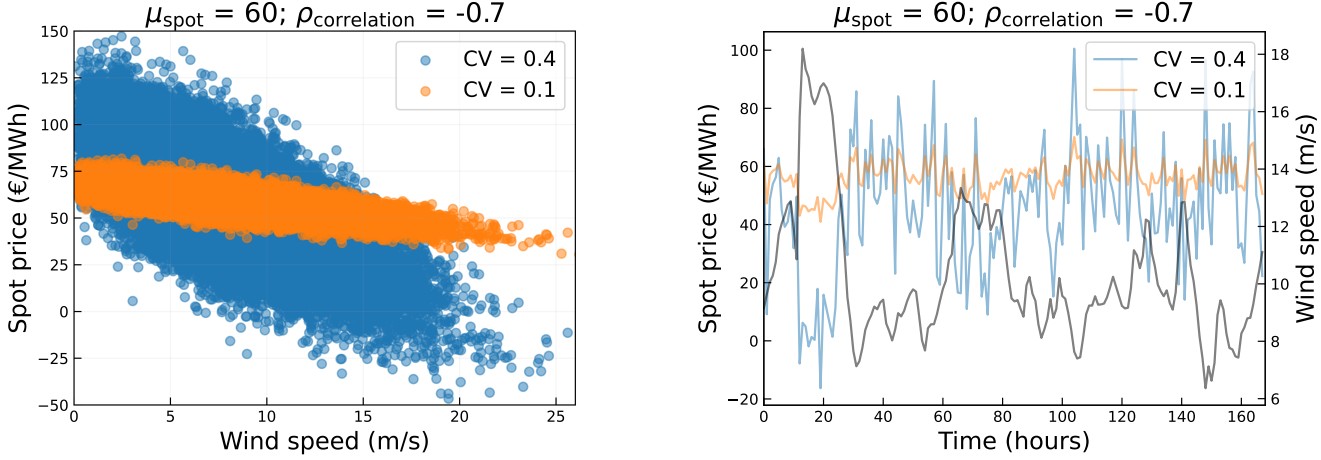

**Figure 4.** Effect of Coefficient of Variation (CV) for a given mean spot price and correlation coefficient

by $\mu_{\text{spot}}$ and CV) such that the correlation between the spot price vector and the input wind speed vector is equal to the defined

$\rho_{\text{correlation}}$. It should be noted that the prices are generated for a year (using hourly wind speed data) and that the corresponding
revenue is considered to be the same for all years throughout the lifetime of the wind farm (corrected for inflation). With the
approximated values of $\mu_{\text{spot}}$, CV, and $\rho_{\text{correlation}}$ for Denmark and the Netherlands, as shown in Fig. 2 and Fig. 3, the spot
prices can be generated, as shown in Fig. 5.

The purpose of the simplified market model is to represent the relation between spot prices and local wind speed with the

help of two defining parameters ($\mu_{\text{spot}}$ and $\rho_{\text{correlation}}$) that can be easily varied to simulate multiple realizations of the future



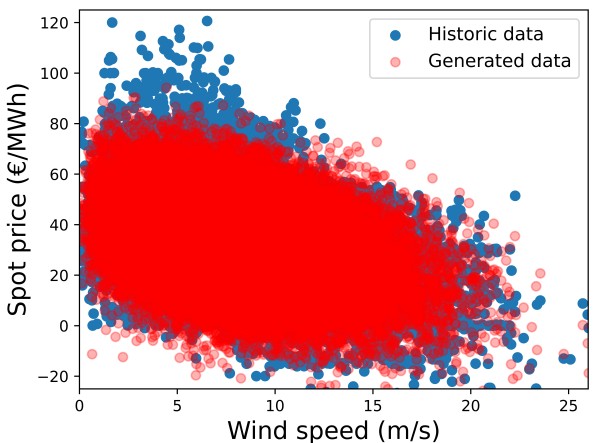
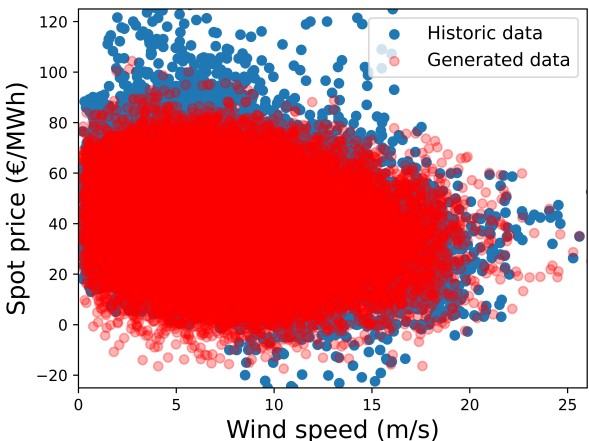

**Figure 5.** Historic spot price data and generated data for Denmark (left) and the Netherlands (right)

market. For instance, a high correlation coefficient would represent a location with a high wind penetration like Denmark. A correlation of zero represents a constant average price per kWh which could be the case with a PPA or a fixed feed-in tariff.

The spot prices are used, eventually, to determine the annual revenue of the wind farm, which will further be used to evaluate the chosen economic objective function. The revenue of a hypothetical 1 GW wind farm in Denmark is calculated using the historic spot price data and the generated spot price data, shown in Fig. 5. Both values are found to be similar, with a difference of less than 1%. A similar difference is observed also for the Netherlands. This implies that although the model misses out on complex market dynamics, the calculated annual revenues are in the right order, making it fit for the purpose of this study.

## 3  Problem formulation

This section discusses the formulation of the optimization problem. The problem formulated is given in Eq. (2) where the objective is to maximize the economic performance of the offshore wind farm w.r.t the rated power ($P$) and rotor diameter ($D$) of the turbine. Maximizing economic performance implies maximizing/minimizing the objective function, depending on the metric used. This is subject to equality constraints w.r.t. the farm rated power ($P_{\text{farm}}$) and the area occupied by the wind farm ($A_{\text{farm}}$).

$$
\begin{aligned}
min/max_{P,D} \quad & \text{f(x)} \\
\text{s.t.} \quad & P_{\text{farm}} = 1\,\text{GW} \\
\text{s.t.} \quad & A_{\text{farm}} = 150\,\text{km}^2
\end{aligned}
\tag{2}
$$

Fig. 6 shows the discrete set of values w.r.t. both the design variables ($P$ and $D$), for which simulations are performed. To match the constant farm power constraint, the number of turbines reduces as the rated power of the turbine goes up, as seen




from the secondary y-axis. A polynomial surface is fit to the data at these discrete points so as to evaluate a property of interest for any given combination of rated power and rotor diameter.

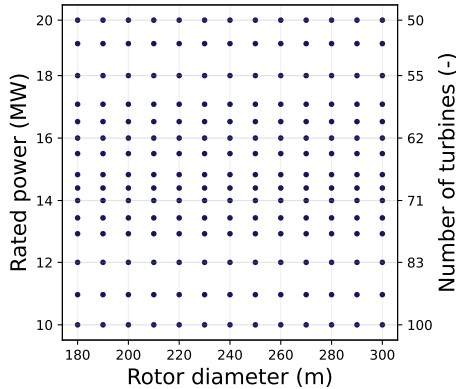

**Figure 6.** Complete design space showing all the combinations of rated power and rotor diameter

The equality constraint w.r.t. the farm power represents a case where for a tendered wind farm, the grid connection is a
given. As a consequence of the farm power constraint, the number of turbines reduces with an increase in the rated power of the turbine. The equality constraint w.r.t. the farm area represents a case where a fixed plot of ocean area is allocated to the developer to build the wind farm. As a result of this constraint, for a regular square layout, the absolute spacing between the turbines depends on the number of turbines that are placed within the given area. These constraints are used for the baseline case as it is assumed to be the most representative of how current commercial wind farms in recent years have been tendered
(Rijkswaterstaat, 2021). The optimum for various market scenarios and objectives, resulting from this formulation, will be compared against the traditional design optimized for minimum LCoE, using the same problem formulation.

## 4   Case study

This section discusses the case study analyzed for the formulated problem. The case study discusses different market scenarios for which several objectives will be maximized/minimized. The site parameters used to carry out all the analyses are also
defined.

### 4.1   Objective functions and market scenarios

This section lists the various economic metrics used as objective functions in the study. The different market scenarios, for which each of these metrics will be evaluated, are also discussed.





### 4.1.1 Objective functions

Various economic metrics exist that are often used to evaluate the profitability of a project. Metrics commonly used for financial assessment of renewable energy projects include IRR, NPV, Benefit-to-cost ratio (BCR), Return on Investment (ROI), simple or discounted Payback Period (PBP), etc. (Delapedra-Silva et al., 2022; de Oliveira et al., 2011). González et al. (2010) optimize the layout of a wind farm using NPV as the objective function. Shamshirband et al. (2014) use both NPV and IRR to optimize the number of turbines to be installed in a wind farm. Ciavarra et al. (2022) optimize the hub height of each turbine in the farm

using both AEP and IRR as objective functions. Joshi et al. (2023) normalize the NPV with the energy output when optimizing airborne wind energy systems. Pookpunt et al. (2020) perform wind farm layout optimization for a variety of metrics including NPV, IRR, and PI. Habbou et al. (2023) look at PI and LCoE to evaluate the profitability of hybrid power plants in European markets. Simpson et al. (2020) proposed CoVE as a metric that normalizes the LCoE with the value factor, which captures the value received by the wind farm developer w.r.t. the average market clearing price.

Each metric has a different formulation and certain benefits/drawbacks. NPV is a measure of the absolute profit of the project since it's a summation of the initial investment and the present value of the future revenues. Since it's not normalized, it is often used to compare the returns of different projects with a similar initial investment. For a design problem where the investment varies with a change in the design variables, the use of NPV can be problematic. For instance, consider that an investment of €10 yields €20 of discounted revenues. Even if simply doubling the investment yields twice the revenue, the NPV would also

double, indicating a much better design which may clearly not be the case. This problem is solved by metrics like PI or BCR, which essentially normalize NPV with the initial investment. However, the use of NPV together with IRR is quite common in capital budgeting and in various academic studies, as exemplified above. IRR is the rate of return at which NPV is zero. Since IRR is normalized and indicates a % return, it serves better for a design problem with varying investments. However, even IRR is to be used with caution. Due to yearly revenues, the invested sum is gradually paid off. Thus, the rate of return is not

achieved over the total lifetime for the total investment. CoVE simplifies cash-flow effects with the same real-interest approach as LCoE while considering the dependency of revenues on wind speed. However, not all market price variations are captured by CoVE. For instance, simply doubling the market prices will double the revenues, giving a different PI, NPV, and IRR, but will result in the same value factor and CoVE.

The turbine optimization problem explored in this study involves a change in investment across all elements of the wind

farm. Hence, using metrics like NPV might give misleading results. Metrics like PI and IRR are clearly better suited for a turbine or farm design optimization problem. However, since the other metrics listed above are commonly used in renewable energy financing and in academic studies, the consequence of using potentially inappropriate metrics will also be explored. The following objective functions are considered in this study:

1. **Levelized Cost of Electricity (LCoE):** This will serve as the baseline objective, which is to be minimized. The LCoE

of the wind farm is given by Eq. (3) where $n$ is a given year, $L$ is the operating lifetime of the wind farm and $r$ is the real discount rate. The numerator contains the capital expenditures ($C_{\mathrm{CAPEX}}$) that are paid initially, the summation of all the annual actualized operation and maintenance costs ($C_{\mathrm{OPEX}}$), and the decommissioning costs paid at the end of the





lifetime ($C_{\text{DECOM}}$) while the denominator contains the summation of the actualized Annual Energy Production (AEP) values.

$$\text{LCoE} = \frac{C_{\text{CAPEX}} + \sum_{n=1}^{L} \frac{C_{\text{OPEX}}}{(1+r)^n} + \frac{C_{\text{DECOM}}}{(1+r)^L}}{\sum_{n=1}^{L} \frac{\text{AEP}}{(1+r)^n}} \qquad (3)$$

2. **Net Present Value (NPV):** NPV is a measure of the absolute profit where all the future revenues and costs have been discounted to represent their value in the present. A positive NPV indicates a profitable investment. Eq. (4) shows the formulation of NPV, where $Cf_n$ represents the cash flows over the years and $C_{\text{CAPEX}}$ represents the initial investment, and $r$ is the discount rate. NPV is an objective that needs to be maximized.

$$\text{NPV} = \sum_{n=1}^{L} \frac{Cf_n}{(1+r)^n} - C_{\text{CAPEX}} \qquad (4)$$

3. **Profitability Index (PI):** PI, same as the Present Value Index (PVI) or BCR, is the ratio between the present value of future cashflows and the present value of the initial investment. It is, in essence, the same as NPV normalized with $C_{\text{CAPEX}}$, as shown in Eq. (5) (de Souza Rangel et al., 2016). An index greater than 1 indicates a profitable scenario. PI is an objective that needs to be maximized.

$$\text{PI} = 1 + \frac{\text{NPV}}{C_{\text{CAPEX}}} \qquad (5)$$

4. **Modified Internal Rate of Return (MIRR):** MIRR is a modified version of the IRR which is the rate at which the NPV of a project is zero. However, IRR assumes that the positive cashflows are reinvested at the IRR instead of the company's cost of capital. MIRR takes this into account and also eliminates the issue of having multiple IRRs. It is an objective that needs to be maximized. It is given by Eq. (6) for a case where the cashflow ($Cf$) is constant throughout the lifetime, $L$, and $r$ is the reinvestment rate of the revenue (de Souza Rangel et al., 2016).

$$(1 + \text{MIRR})^L = \frac{Cf}{C_{\text{CAPEX}}} \left[ \frac{(1+r)^L - 1}{r} \right] \qquad (6)$$

5. **Cost of Valued Energy (CoVE):** CoVE, proposed by Simpson et al. (2020) covers both the costs and revenue aspects. It is a function of the LCoE of the farm and the value factor (VF), as shown in Eq. (7). CoVE is an objective that needs to be minimized.

$$\text{CoVE} = \frac{\text{LCoE}}{\text{VF}} \qquad (7)$$





The value factor for a specific year is given by Eq. (8) where the average price that the wind developer receives is normalized with the mean spot price.

$$\text{VF} = \frac{\frac{\sum_t P_{\text{farm}} \cdot \lambda_{\text{spot}}}{\sum_t P_{\text{farm}}}}{\mu_{\text{spot}}} \tag{8}$$

### 4.1.2 Market scenarios

The future spot prices are highly uncertain and it is difficult to make a prediction of the same. This study does not aim at making any future price predictions. Instead, various scenarios are simulated wherein the essential parameters of the market model, the mean spot price ($\mu_{\text{spot}}$), and the correlation coefficient ($\rho_{\text{correlation}}$) are varied. The variations in the mean and correlation parameters result in a differing behavior of spot prices w.r.t. the site-specific wind speed. A mean spot price of around 40 €/MWh is already common for both Denmark and the Netherlands, with some years having a mean of up to 60 €/MWh. Due
to the high wind energy penetration in the grid, some offshore sites in Denmark experience a correlation of about -0.4. The years from 2020-2022 have been unusual and have led to some unforeseen price spikes. For instance, the prices skyrocketed across Europe in 2022, with the Netherlands experiencing a mean spot price of about 250 €/MWh. These years are anomalies and cannot be treated as representative data points. Gonzalez-Aparicio et al. (2013) predict a mean price of about 55 €/MWh for the Dutch market by 2030 in a high electrification scenario, while Swamy et al. (2022) predict an average price of about
51 €/MWh by 2030 and 104 €/MWh in 2050. However, the prediction of future spot prices largely depends on the inputs and the assumptions of how the capacities of various technologies and prices (of fuel, carbon, etc.) evolve in the future. Hence, this study uses a range of 40 €/MWh to 100 €/MWh for the mean spot price. For the correlation coefficient, the entire range from 0 (no correlation) to -1 (perfectly anti-correlated) is explored. As mentioned before, the coefficient of variation is fixed at a value of 0.4. These parameters will differ per region. In regions with a high wind penetration, high correlation coefficients
can be expected whereas, in regions with a low penetration, wind generation might not have a strong influence on spot prices resulting in a lower correlation between spot prices and wind. Also, several factors other than the technology mix, like demand response, number of electric vehicles in the system, electrolyzer penetration, etc., will determine the correlation coefficient, and the mean and the coefficient of variation of the spot price distribution.

The range of values, along with the number of grid points used in the given range, is shown in Table 1. The values at
the bounds will result in extreme optimums and a change in these bounds will simply shift these extreme optimum designs. However, instead of the boundary points of the input, what is interesting is how the mean price and the correlation coefficient drive the optimum. Each combination of these two parameters represents a particular market scenario, with a total of 154 market scenarios being simulated.



**Table 1.** Market parameter variations

| Parameter | Range | No. grid points | Unit |
|---|---|---|---|
| Mean spot price ($\mu_{\text{spot}}$) | [40, 100] | 14 | €/MWh |
| Correlation coefficient ($\rho_{\text{correlation}}$) | [-1, 0] | 11 | – |

## 4.2 Site parameters

In this study, a hypothetical site and wind farm typical for the North Sea are considered. The site parameters and the farm orientation define the case study. Fig. 7a shows the wind rose for the hypothetical site with the highest probability of all wind speeds occurring along the South-West direction.

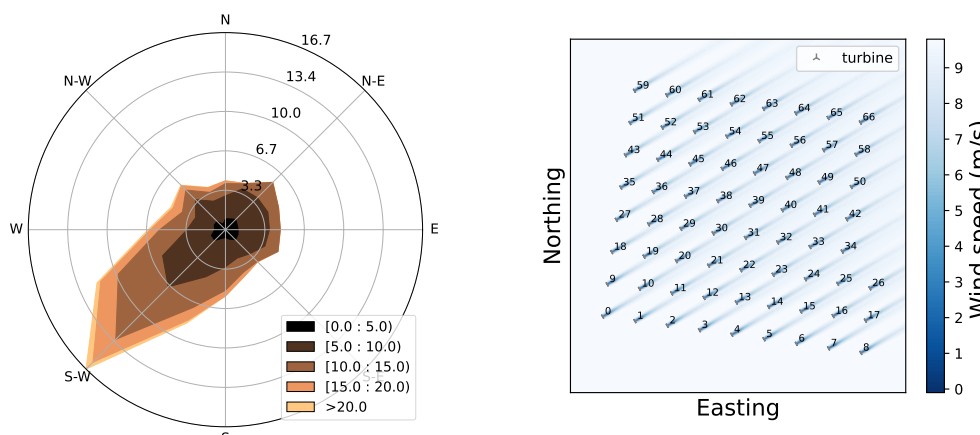

**Figure 7.** (a) Directional wind speeds and probabilities for the hypothetical site (b) Farm layout for a 15 MW turbine (and 1 GW of farm power)

Some other case-defining parameters, like the distance to grid, water depth, etc., are listed in Table 2. The mean wind speed mentioned is at 100 m and is always projected to the hub height of the turbine design being analyzed using the power law.

**Table 2.** Case study parameters

| Parameter | Value | Unit |
|---|---|---|
| Distance to grid | 60 | km |
| Distance to harbor | 40 | km |
| Water depth | 30 | m |
| Mean wind speed at 100 m | 9.4 | ms$^{-1}$ |
| Maximum wave height (50 year) | 5 | m |
| Wind farm lifetime | 25 | years |





Fig. 7b shows the farm layout for the reference turbine, along with wind speed deficits for a given wind speed, coming from the South-West direction. Given the farm power constraint, a farm power of about 1 GW and a turbine rated power of 15 MW result in around 67 turbines. The layout is always close to a regular square layout. Hence, the turbines are first arranged in a square grid of 64 turbines and the remaining 3 turbines are added along a new column. Since there is also a farm area equality constraint, the turbines are always forced to use all the farm area ($150 \ \mathrm{km}^2$). Hence, for a given number of turbines, the absolute
distance between the turbines is fixed and the normalized spacing depends on the rotor diameter.

## 5    Results

This section first discusses the resulting optimum for all the different scenarios. The differences in performance, for each metric, across the entire design space are then shown for a given market scenario. Finally, the overall performance of a few designs across all the market scenarios and objective functions is discussed.

### 5.1    Optimum designs for all market scenarios

For each market scenario, the optimum design may result in a positive business case or a negative business case. For instance, a positive business case has a profitability index higher than 1, a MIRR larger than the discount rate used for LCoE, and a positive NPV. A negative business case has a profitability index of less than 1, a MIRR lower than the discount rate used for LCoE, and a negative NPV. It is important to understand that the lowest specific power designs (low ratings and larger
rotors) have the steepest power curve and the highest AEP and revenue, while the designs with the highest specific power (high ratings and smaller rotors) have the lowest wind farm costs. Fig. 8a shows the gradients for the total costs over the entire design space, where the gradients always point towards the turbine with the highest rating and lowest rotor diameter. This is because an increase in rating decreases the number of turbines in the farm, reducing the O&M costs and installation costs, and a decrease in the rotor diameter decreases the turbine and support structure costs. Fig. 8b shows the revenue gradients from
selling electricity in the spot market, for a given market scenario. It can be seen how the gradients point towards the turbine with the lowest rating and the largest rotor, resulting in the steepest power curve, having the highest revenue. Higher rated power in itself doesn't lead to higher revenues, since the total power of the farm remains constant. When changing the market scenario, the cost gradient for all the designs remains unchanged, while the revenue gradient for a design over its lifetime is altered. These gradients are shown to support later interpretations and explanations of the results.
Fig. 9a shows the LCoE of the entire design space along with the global optimum (rated power of 16 MW and rotor diameter of 236 m) that is already close to some of the state-of-the-art turbines, while Fig. 9b shows the optimum designs for all the market scenarios and all economic metrics. This optimum for LCoE serves as the baseline for comparison against market-optimized designs. A detailed discussion about the LCoE-optimum and its sensitivity to various model parameters, design inputs, and the problem formulation can be found in Mehta et al. (2023).
For each metric, the optimum designs for various market scenarios are different and are separately plotted. The optimum designs for market scenarios that resulted in a negative business case are plotted with a high transparency. Various market




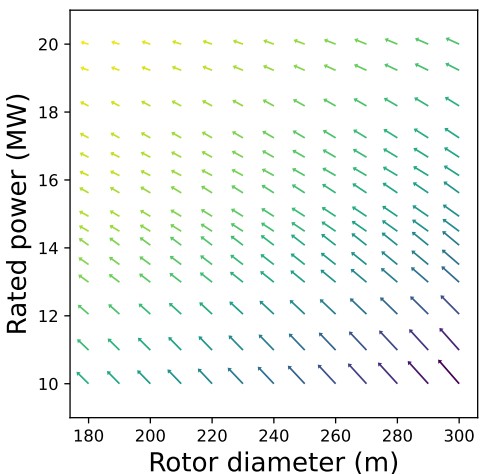 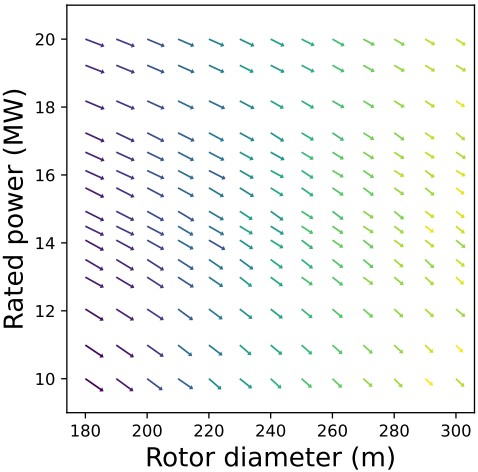

**Figure 8.** (a) Cost gradients for the entire design space (b) Revenue gradients for the entire design space

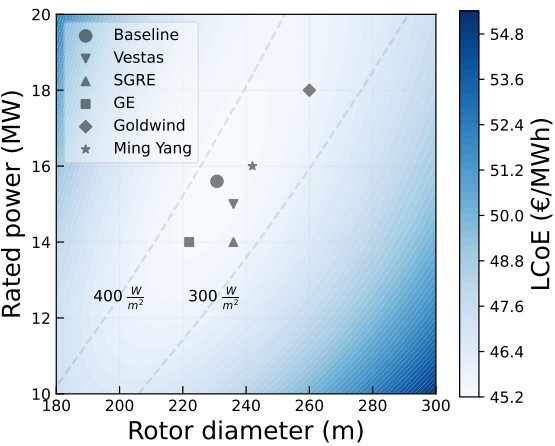 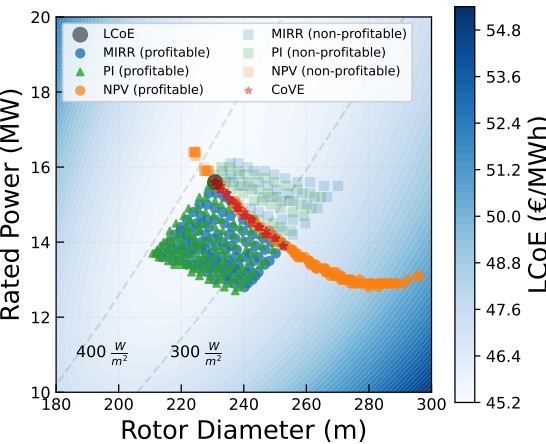

**Figure 9.** (a) Optimum turbine design for LCoE (b) Optimum turbine design for various objective functions and market scenarios

scenarios result in a spread of optimum designs for both MIRR and PI, while for NPV and CoVE, the optimums always move in the same direction. For NPV, the optimum approaches the rotor diameter limit at high mean spot prices. This can be attributed to the behavior of the cost and revenue gradients, especially closer to the boundaries. It can be seen that depending
on the choice of the economic metric and the realization of the future market, the optimum can differ significantly compared to the traditional LCoE-optimized design. Further explanations on how different market model parameters ($\mu_{\text{spot}}$ and $\rho_{\text{correlation}}$) drive the optimum for each economic metric are given below.



### 5.1.1 Effect of mean spot price

The mean spot price has a different effect on each metric. A change in the mean spot price also changes the standard deviation
(as CV is constant) and hence, the distribution from which the prices are sampled. Fig. 10 shows how, for a $\rho_{\text{correlation}}$ of 0,
the mean spot price drives the optimum in different directions, depending on the metric. The arrows in the figure point in the
direction of increasing mean spot prices.

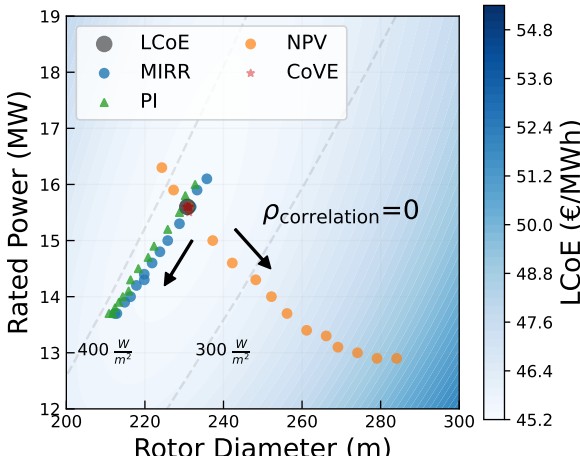

**Figure 10.** Optimum designs for different mean spot prices (for a correlation coefficient of 0)

The effect of the mean on the optimum for different metrics can be better explained by looking at the formulation of the
metric itself. CoVE depends on the LCoE and the value factor (shown in Eq. (8)). The value factor is the ratio of the price
received by the developer (wind farm power-weighted average of the spot prices) and the mean spot price. An increase in the
mean spot price almost equally increases the received spot price by the developer, canceling out the effect. Hence, a change in
mean spot price has an insignificant effect on the value factor. As a consequence, CoVE and the optimum design w.r.t. CoVE
do not change with a change in the mean spot price. MIRR and PI are metrics that are normalized with the initial investment
and exhibit a similar behavior w.r.t. the shift in optimum. It's a measure of the best return (revenue) per euro invested. The
behavior of both metrics can be explained by looking at the formulation of PI, as shown in Eq. (9), where the cashflow in each
year $(Cf_n)$ is the net revenue, which is simply the operations and maintenance costs $(C_{\text{O\&M}})$ taken out from the total revenue
earned from selling the electricity in the spot market $(R_{\text{spot}})$.

$$\text{PI} = \frac{\sum_{n=1}^{L} \frac{Cf_n}{(1+r)^n}}{C_{\text{CAPEX}}}$$

$$= \frac{\sum_{n=1}^{L} \frac{R_{\text{spot}}}{(1+r)^n} - \sum_{n=1}^{L} \frac{C_{\text{O\&M}}}{(1+r)^n}}{C_{\text{CAPEX}}} \tag{9}$$



For simplification, the summation of discounted revenues is written as R, and the summation of discounted operations and maintenance costs is written as O. The gradient of PI w.r.t. the rotor diameter (D) can be given by Eq. (10). The gradients, along with their associated weights, for the revenue, operation and maintenance costs, and the initial investment are clearly separated. The gradient w.r.t. the rated power (P) can be similarly calculated.

$$\frac{\partial \mathrm{PI}}{\partial D} = \frac{\partial}{\partial D}\left(\frac{\mathrm{R}-\mathrm{O}}{C_{\mathrm{CAPEX}}}\right)$$

$$= \frac{1}{C_{\mathrm{CAPEX}}}\cdot\frac{\partial \mathrm{R}}{\partial D} - \frac{1}{C_{\mathrm{CAPEX}}}\cdot\frac{\partial \mathrm{O}}{\partial D} - \frac{(\mathrm{R}-\mathrm{O})}{C_{\mathrm{CAPEX}}^2}\cdot\frac{\partial C_{\mathrm{CAPEX}}}{\partial D} \tag{10}$$

A change in the market scenario directly affects the revenue gradient but also results in a different absolute revenue. Hence, the weight of the gradient for $C_{\mathrm{CAPEX}}$ also changes with a change in the market scenario. The effect of the mean spot price on these gradients will determine the direction in which the optimum is driven.

Since NPV is simply a summation of the discounted revenues, initial investment, and discounted operations and maintenance costs, the gradients of NPV w.r.t. the rotor diameter are given by Eq. (11). A change in the market scenario only alters the magnitude of the revenue gradient, while the cost gradients remain unaffected, unlike for PI or MIRR.

$$\frac{\partial \mathrm{NPV}}{\partial D} = \frac{\partial \mathrm{R}}{\partial D} - \frac{\partial \mathrm{O}}{\partial D} - \frac{\partial C_{\mathrm{CAPEX}}}{\partial D} \tag{11}$$

The effect of mean spot price on the gradients for NPV and PI is shown in Fig. 11, where the gradients at the LCoE-optimized design are plotted. The gradients for a mean spot price of 45 €/MWh and a mean spot price of 100 €/MWh, both with a correlation of 0, are shown. The gradients are normalized with the magnitude of the gradient with the maximum value.

A market scenario with a mean price of 45 €/MWh and a correlation of 0 represents a scenario where a fixed price around the minimum LCoE value is received for every unit of energy produced. The summation of the gradient (with the weights) for the initial investment ($C'_{\mathrm{CAPEX}}$) and the gradient (with the weights) for the operations and maintenance costs (Ó) result in the total cost gradient ($C'_{\mathrm{total}}$). For the mean spot price of 45 €/MWh (and correlation of 0), the gradients for costs and revenue are in balance, indicating that the LCoE-optimum point is also the market-optimum design, for both metrics.

An increase in the mean spot price to 100 €/MWh clearly has an impact on the revenue gradient (Ŕ). This is the only change in the NPV gradients. Hence, the optimum moves along the direction of the revenue gradient. Any change in the mean spot price will always move the optimum along the direction of the revenue gradient, which was shown in Fig. 8. For the gradients of PI, shown in Fig. 11 (right), it can be seen that, since the absolute revenue also changes, the weight of $C'_{\mathrm{CAPEX}}$ also goes up (shown in Eq. (10)). This results in an increase in $C'_{\mathrm{CAPEX}}$. Since Ó does not change, a change in $C'_{\mathrm{CAPEX}}$ causes a shift in both magnitude and direction of the total cost gradient ($C'_{\mathrm{total}}$). This change in direction leads to a shift in optimum along a different direction than the direction along the line of $C'_{\mathrm{total}}$ and Ŕ. The resultant of the new cost and revenue gradients pushes the optimum towards downsized turbines with lower ratings and smaller rotors. The same effect is also observed for MIRR.





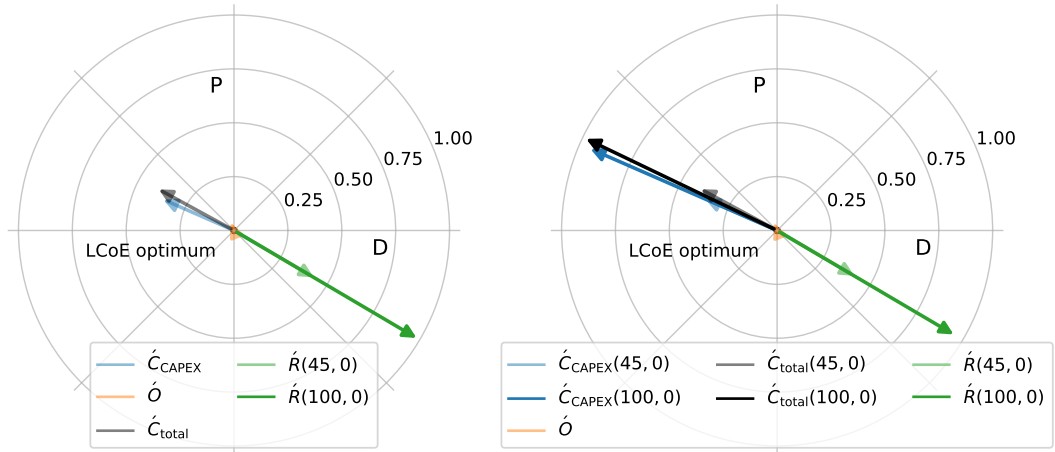

**Figure 11.** Effect of change in mean spot price on NPV gradients (left) and PI gradients (right) at the LCoE optimum

This is represented by the optimum designs along the constant specific power line, shown in Fig. 9b. From the LCoE-optimum design, the optimum designs in the direction of the constant specific power are driven by different mean spot prices. That line corresponds to a correlation of 0. Similarly, for other values of the correlation coefficient, the mean spot price also drives the optimum in the direction of constant specific power, albeit from a different starting point than the LCoE optimum.

### 5.1.2 Effect of the correlation coefficient

The correlation coefficient affects the rate of change of spot prices w.r.t. the wind speed. For the same mean and standard deviation, a correlation of zero results in no relation between the spot prices and wind speed, while a correlation of -1 results in a perfectly anti-correlated line. However, the correlation coefficient drives the optimum differently, compared to the mean spot prices. Fig. 12 shows how, for a fixed $\mu_{\text{spot}}$ of 45 €/MWh, the correlation coefficient drives the optimums in the same direction but with differing magnitudes, depending on the metric. The arrows in the figure point in the direction of increasing (more negative) correlation coefficients.

The effect of change in the correlation coefficient on the gradients for NPV and PI is shown in Fig. 13. The figure shows the revenue and cost gradients for a market with a mean price of 45 €/MWh and for two different correlation values, 0 and -1, resembling no correlation and perfect anti-correlation. It can be seen that the revenue gradient (Ŕ) for the high correlation case is slightly larger than the gradient with no correlation. For NPV, that is the only change in the gradients, again driving the optimum along the direction of the revenue gradient. However, it can be seen that the change in the revenue gradient is insignificant compared to the change caused by the variations in the mean spot price.

For the same mean spot price, an increase in the correlation coefficient increases the value of the power produced at lower wind speeds and results in a shift in optimum towards larger rotors (lower specific powers). For PI (and MIRR), because of the change in the absolute revenue, the weight of $C'_{\text{CAPEX}}$ decreases, as indicated by the decrease in the vector magnitude.



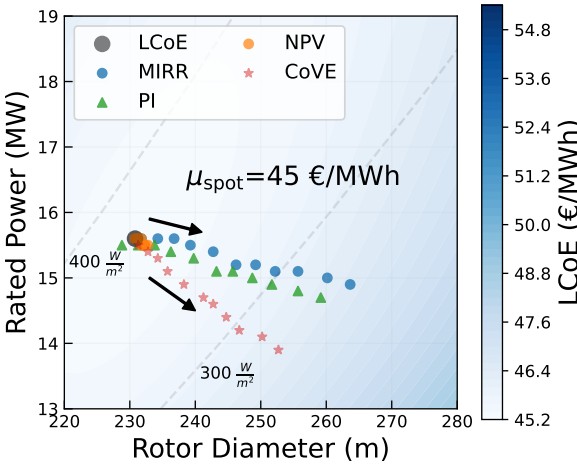

**Figure 12.** Optimum designs for different correlation coefficients (for a mean spot price of 45 €/MWh)

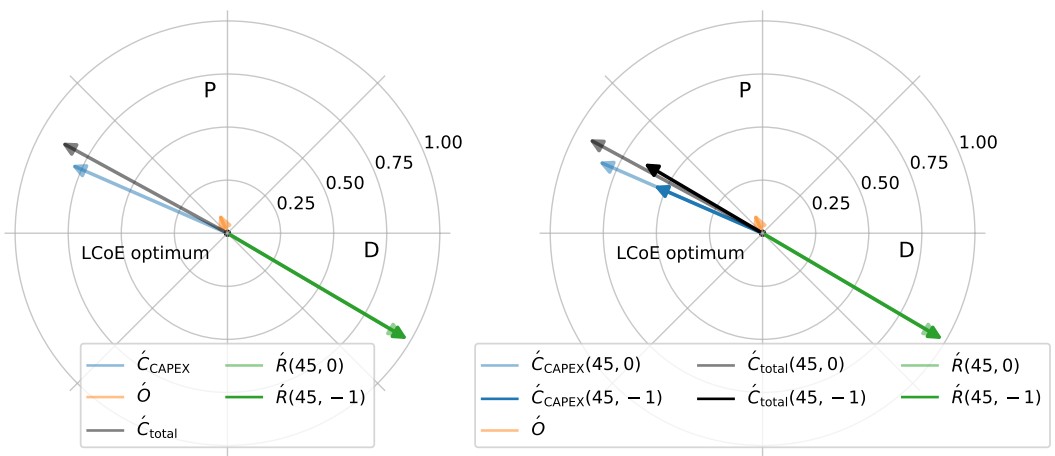

**Figure 13.** Effect of change in the correlation coefficient on NPV gradients (left) and PI gradients (right) at the LCoE optimum

This also causes a slight change in the direction of the total cost gradient ($C'_{\text{total}}$). For the scenario with a high correlation, the magnitude of Ŕ is much larger compared to the magnitude of $C'_{\text{total}}$. The resultant of these two vectors drives the optimum in the direction of the revenue gradient, same as for NPV. The difference in magnitude of the two vectors for PI and MIRR is much larger than the difference observed for NPV. Hence, the correlation has a significantly larger effect on the optimum design w.r.t. PI and MIRR, compared to NPV.

For higher correlation coefficients (more negative), the power produced at lower wind speeds is valued much more than at higher wind speeds. Hence, designs with a low specific power have a higher value factor for market scenarios with high (more





negative) correlation coefficients. As a consequence, the correlation coefficient also drives the optimum w.r.t. CoVE towards
larger rotors and lower ratings.

### 5.1.3    Summary of the effect of market parameters

This section summarizes how both the market parameters drive the optimum turbine design. The results show that the choice of
metric has a crucial impact on the magnitudes and directions of changes in the optimal designs w.r.t. changes in the mean spot
price and the correlation coefficient. For NPV, only the magnitude of the revenue gradient changes when the mean spot price
or the correlation coefficient changes. For PI/MIRR, along with the changes in the magnitude of the revenue gradient, both the
magnitude and direction of the cost gradient also change. This is caused by the effect that normalization by mean revenues has
on the weights of the CAPEX gradient.

For changes in the mean spot price, absolute profits (NPV) drive the solution in a direction perpendicular to the change
for normalized profits (PI, MIRR). The magnitude of change is significant for both NPV and PI/MIRR. Normalization with
the mean revenue, for CoVE, makes the design insensitive to changes in the mean spot price. For changes in the correlation
coefficient, the direction of change in the optimum does not depend on the metric, and the optimum is always driven in the
direction of changing specific power. A larger (more negative) correlation pushes the optimum towards larger rotors and lower
power ratings (lower specific powers). However, the magnitude of change is significantly larger for PI/MIRR than that for NPV.
Since only the correlation coefficient influences the value factor, the spread of optimum designs for CoVE, shown in Fig. 9b,
can be attributed to changes in the correlation coefficient.

The change in the gradients of each metric explains how the optimum shifts w.r.t. changes in the mean spot price and the
correlation coefficient. Section 5.1.1 and Section 5.1.2 also show how metrics other than MIRR or PI may lead to completely
different design trends, especially w.r.t. the mean spot price. As discussed before, the difference in behavior is a result of the
formulation of the metric itself. It is, now, also apparent how the results of NPV differ and might be misleading, compared to
other economic metrics, for optimization problems with changing investments. Although CoVE captures the changes in the
market value of wind, it does not respond to changes in the mean spot prices, and hence, may not be ideal when evaluating
the business case for a developer. Therefore, in the following sections, only the analyses w.r.t. PI and MIRR are discussed in
further detail.

### 5.2    Performance of all designs for a single market scenario

In the previous section, Fig. 9a showed the LCoE across the entire design space. It also showed that the LCoE along the constant
specific power line does not change significantly, compared to the LCoE at the optimum, even for large changes in the design.
Similarly, even though the market scenarios result in different optimum designs, it is important to evaluate the difference in the
absolute performance of PI and MIRR across the entire design space. A market scenario that results in a large change in the
optimum, compared to the LCoE-optimized design, is considered. The performance across the entire design space for a market
scenario with a mean price of 100 €/MWh and a correlation coefficient of -1 is shown in Fig. 14. For both PI and MIRR, the
global optimum for the given market scenario and the LCoE optimum are also shown.





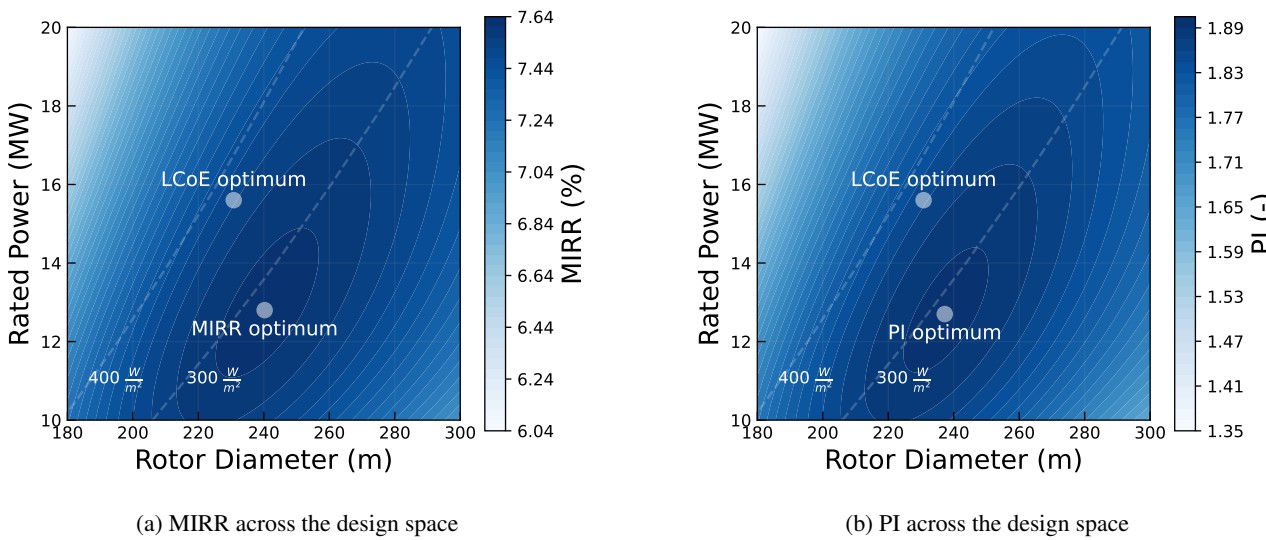

(a) MIRR across the design space

(b) PI across the design space

**Figure 14.** Performance of the entire design space w.r.t. MIRR and PI for a market scenario with $\mu_{\text{spot}}$ = 100 €/MWh and $\rho_{\text{correlation}}$ = -1

For both MIRR and PI, it can be seen that although the market-driven optimum is different from the LCoE optimum, the difference in the value of the metric itself is insignificant. The value of MIRR and PI for the LCoE-optimized design is about 2-3% lower than the maximum value of the design specifically optimized for MIRR and PI. It can be seen that even for an extreme market scenario, the values for MIRR and PI for a large range of designs around the optimum are similar to the value for the optimum design. Depending on the market scenario, the optimum differs and so does the difference in the absolute value of the metric. However, the difference in the absolute values of MIRR and PI itself is less significant.

### 5.3 Performance of different designs over all market scenarios

It is clear that for any given objective, different market scenarios result in different optimum designs. The difference in the value of the metric itself for one market scenario was also discussed in Section 5.2. However, it is crucial to understand if there is any added value in optimizing designs specifically for a certain market scenario and to understand the risk of designing for the wrong market. Hence, the performance of some designs over the complete range of market scenarios is determined for both MIRR and PI. The designs used for comparison are the LCoE-optimized design (16MW-236m), a downsized turbine (14MW-220m) with similar specific power as the LCoE-optimized turbine, a low specific power turbine (14MW-260m), and an upscaled turbine close to the state of the art in the industry (18MW-260m), but with slightly lower specific power than the LCoE-optimized turbine. Fig. 15 shows the performance of the four sample designs plotted against $\mu_{\text{spot}}$ for the two extreme values of $\rho_{\text{correlation}}$. The dotted horizontal line in the plots separates the profitable and non-profitable values.

Clearly, both MIRR and PI increase with an increase in the mean spot price, for all the designs. Also, the values drop with an increase in the correlation coefficient (more negative). This is simply because a higher mean spot price (for the same correlation) results in higher revenues and a higher correlation (for the same mean spot price) leads to lower prices at high-





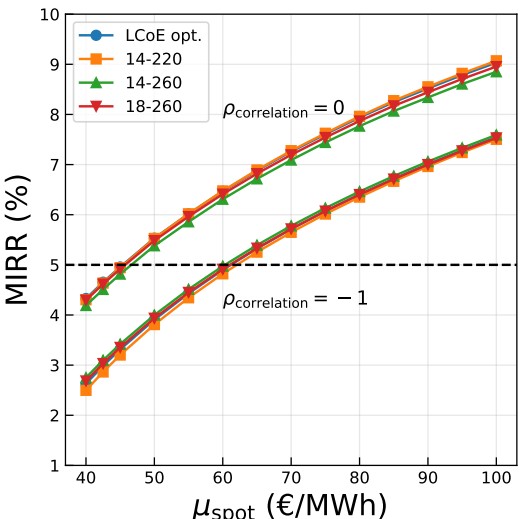
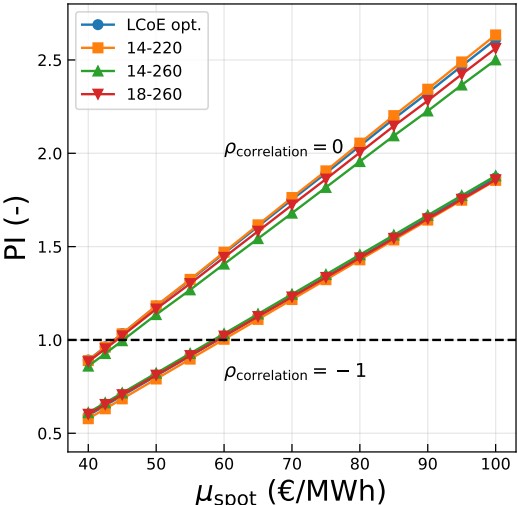

**Figure 15.** Performance of the designs w.r.t. MIRR (left) and PI (right) over the complete range of market scenarios

yield wind speeds, resulting in lower revenues. It can also be seen that the variations in the economic value due to the design choices are insignificant compared to the variations due to the uncertainties in the market scenario. The mean spot price and the correlation will be determined by how wind generation, demand, and various other technologies develop in the future. These factors will largely determine the economic performance of the wind farm rather than the choice of turbine design. The

485 differences in the performance are significant only when designing for certain extreme market scenarios (low mean spot price and a high negative correlation or high mean spot price and no correlation).

For most scenarios, all the designs exhibit a similar performance, for both MIRR and PI. For a correlation coefficient of zero, the LCoE-optimized design performs better than the low specific power designs, for any given mean spot price. For scenarios with a high correlation, the low specific power designs perform marginally better than the LCoE-optimized design, for any

given mean spot price. This suggests that although a design optimized for the market might have a slightly higher MIRR or PI, an LCoE-optimized design already performs quite well w.r.t. MIRR and PI.

## 6 Conclusions

This research looked at how various economic metrics (MIRR, PI, NPV, and CoVE) and different future market scenarios would drive the optimum turbine design. The research specifically considered turbines in a hypothetical offshore wind farm

in the North Sea where the farm power and area were kept constant. Also, the revenues only from the day ahead market were considered. Some general insights from this study are listed below.

– MIRR and PI exhibit a similar behavior w.r.t. both changes in the mean spot price and the drop in spot prices w.r.t. the wind speed (cannibalization effect). Compared to the LCoE-optimized turbine, an increase in the mean spot price drives



the optimum towards downsized turbines with similar specific power. For regions with a high wind penetration, resulting
in a larger drop in spot prices w.r.t. the wind speed, the optimum shifts towards lower specific power turbines in the
direction perpendicular to the constant specific power line. The study also showed how MIRR and PI, which normalize
the revenues with the initial investment, are better suited for a turbine optimization problem compared to NPV, which
measures absolute profits.

– The benefits of redesigning the turbine for a specific market scenario are marginal. It is seen that even in an extreme
market scenario, the values of MIRR and PI for an LCoE-optimized are 7.5% and 1.86 respectively, while the values for
the market-optimized design are 7.6% and 1.89, respectively. The relative differences are insignificant for most market
scenarios. However, for a market scenario with very low prices, the relative differences between the designs are higher.

– The impact of the choice of the design itself on MIRR/PI is found to be insignificant for most market scenarios. The value
of MIRR/PI for most designs in a wide range of specific powers (200-400 $\mathrm{Wm}^{-2}$) is only up to 10% lower compared to
the value of the market-optimized design.

To operate in future subsidy-free day-ahead markets, the optimum and economic performance will largely be governed by
how market prices develop. However, for metrics like MIRR and PI that allow a fair comparison of designs, a large range of
designs perform well. The results of this study indicate that there is a limited need to focus efforts on redesigning turbines that
are better suited for a specific market scenario. LCoE-optimized turbines are found to perform well for most day-ahead market
scenarios. Turbine optimization might still be largely driven by various other factors like wind resources, farm parameters, grid
and/or area constraints, etc.

*Code availability.* The code along with all the input files and plot scripts is open source and is available at:
https://doi.org/10.5281/zenodo.8380355

*Author contributions.* **Mihir Mehta:** Conceptualization, Data curation, Formal analysis, Investigation, Methodology, Project administration,
Software, Validation, Visualization, Writing - original draft, Writing - review and editing. **Michiel Zaaijer:** Conceptualization, Methodology,
Software, Validation, Visualization, Supervision, Writing - original draft, Writing - review and editing. **Dominic von Terzi:** Conceptualiza-
tion, Methodology, Validation, Visualization, Supervision, Writing - review and editing.

*Competing interests.* The authors declare that they do not have conflicts of interest.



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
