# Peer review of "Designing wind turbines for profitability in the day-ahead markets"

_Wind Energy Science, 2024_

## Author Comment (AC1)

Manuscript ID: WES-2024-43

Designing wind turbines for profitability in the day-ahead markets

July 15, 2024

The authors would like to thank the reviewers for examining the revised manuscript and for their valuable additional feedback. The reply to the comment made by the reviewers is marked in blue, and the literal changes in the paper are marked in red.

**Reviewer 1 comment**

Mehta et al., 2023 investigated the drivers for sizing offshore wind turbines when minimising the LCoE. In this follow-up paper, the authors explore how different day-ahead market price scenarios affect the turbine sizing and compare it to the LCoE-optimised design.

Both papers use the same MDAO modelling framework. It consists of a chain of low-fidelity models aiming to capture the combined effect of two wind turbine (WT) design variables: rated power and rotor diameter. Finding the optimal design requires the definition of an optimisation objective and wind farm constraints. The new paper defines objective metrics that include the day-ahead market price but applies the same wind farm constraints to all investigations: rated power (Pfarm=1 GW) and area (Afarm=150 km2). While the paper provides valuable insights into market value objectives, keeping the wind farm constraints fixed raises questions about the study's validity. Denmark's new 6 GW offshore tender has a per GW area of about 450 km2, or about three times that of the current work.

The authors conclude that the benefits of making market-optimised WTs are minimal - the LCoE-optimised WT performs similarly to the market-optimised. However, I am unsure if that conclusion is not due to the wind farm (WF) constraints. The purpose of redesigning the WT, e.g., by increasing its rotor area, is to increase production at lower wind speeds where electricity prices are higher. Similarly, decreasing the WT's rated power reduces costs while only decreasing low-value output at high wind speeds. Conceptually, the authors optimise the WT's power curve to new market conditions. However, with the fixed constraints on WF area and rated power, the resulting WF power curve is locked irrespectively of the chosen WT design. The placement of the wind turbines further enforces this fixation. The turbines are aligned towards the main wind direction, i.e. they are placed to maximise production, not to optimise market value. Like the WF area, the regular rectangular layout does not represent current commercial wind farms.

The paper provides a good overview of objective metrics, including the day-ahead market value of wind energy. It also shows how changing the WT size influences these metrics for a single wind farm case. However, with the new market incentives, the authors should adapt both the design objectives and the constraints. The paper should provide other examples with WF constraints and layouts representing modern wind farms. The authors can do this by removing the wind farm power constraints, changing the wind farm area and optimising the WF layout for market value instead of AEP. As it is, only a single case study is presented— with very restrictive constraints making it hard to draw the general conclusions that they are.

The reviewer raises an interesting point about the consequences of using the constraint values and the constraint formulation. In the first paper that focused explicitly on LCoE optimization (https: //wes.copernicus.org/articles/9/141/2024/wes-9-141-2024.pdf), the shift in optimum because

of both constraint values and the formulation is explored in detail. However, the purpose was to have the constraints representative of current and planned tenders in the North Sea, where most wind farms have a farm power density of 5 to 10 MW/km2 (`https://www.noordzeeloket.nl/en/functions-and-use/offshore-wind-energy/free-passage-shared-use/borssele-wind-farm-zone/`; `https://english.rvo.nl/topics/offshore-wind-energy/wind-farm-zone`). From a spatial planning perspective, a similar farm power density is also expected of other regions with shallow waters and high wind resource to use the limited area efficiently. Moreover, even a large tender like the 6GW tender in Denmark is expected to be split into smaller wind farms, making the constraint values used in the paper still relevant. We couldn't find a direct reference to the $450 \text{km}^2$ area per GW, but a farm power density of about $2\text{MW/km}^2$ might be too low, especially for the North Sea. Hence, the constraint values used in the study also seem relevant for the near future, if not for all tenders, then at least for many of them.

A change in the constraint values is expected to shift the absolute values of the LCoE and market-driven optimum. However, the differences in design between the two approaches will remain similar. The underlying mechanics that drive the differences between an LCoE-optimized and market-optimized optimum are still captured with the current constraint formulation. The two mechanisms described by the reviewer, favoring larger rotors and lower rated powers, are indeed correct and visible in the results of, for instance, Figure 14 of the paper. The authors would also like to emphasize that in both papers, the absolute value of optimum per se is not the focus but rather the drivers for optimum turbine sizing (in paper 1) and the differences between an LCoE and a market-driven approach (in paper 2).

The authors disagree with the statement from the reviewer that the wind farm power curve is locked by the constraints, irrespective of the chosen WT design. A change in the specific power of the turbine causes differences in the farm power curve, eventually leading to different revenues depending on the relation between the spot prices and the wind speeds. This mechanism is captured with the existing setup. Only altering the rotor diameter (keeping the rated power of the turbine constant) changes the farm power curve drastically, mainly because of a change in the partial load region of the turbine power curve itself Fig. 1 (left). On the other hand, changing the turbine's rated power (for the same rotor diameter) mainly alters the number of turbines in the farm while extending the partial load region of the turbine. This extension of the partial load region to higher wind speeds is also visible in the farm power curve, as seen in Fig. 1 (right). A lower farm power for the 20MW turbine (with the same rotor diameter as the 10MW turbine) can be attributed to the reduced number of turbines due to the farm power constraint.

[Figure]

Figure 1: Farm power curve variation due to a change in the turbine rotor diameter (left) and turbine rated power (right)

Both wind farm power curves also include the effect of a change in the magnitude of the wake

losses, but those are secondary to the fundamental change described above. The authors agree that the alignment with the main wind direction aims at maximizing production, but it does this irrespective of wind speed. Therefore, this choice is no more favorable for the LCOE optimized turbine than for the market-driven turbine. The authors also agree that the regular rectangular layout does not represent current commercial wind farms. However, also here, it is not expected that this affects the LCOE optimized turbine and the market-driven turbine in significantly different ways. The alignment is, in general, unfavourable for both designs. Since most wake losses occur just below rated wind speed, the LCOE-driven turbine could be slightly more disadvantaged by it than the market-driven turbine. However, the effect is marginal, and the authors are not aware of any literature that shows that layouts for LCOE maximization and for market-driven optimization differ significantly.

A change in the turbine design should reflect the changes in the farm power curve and farm-level costs. When coupled with the market price module, these essential elements lead to differences between an LCoE-optimized turbine and a market-optimized turbine for a given objective. Since these essential elements are well captured in the current methodology, removing certain constraints or introducing layout optimization might not give additional insights about the differences between an LCoE-driven and a market-driven design. To substantiate the point, preliminary results for a case without the farm power constraint (and only farm area constraint), as suggested by the reviewer, are shown for a few metrics in Fig. 2. The difference between the optimum designs for revenue-based metrics and LCoE is similar to those observed for the baseline case (with both area and power constraints). Also, for an extreme market scenario with a high mean and high correlation, the values for MIRR are shown in Fig. 2 (right). As seen, even though the optimums differ significantly, the value of the metric itself is 7.33% and 7.54%. These results, in terms of the difference in designs and the value of the metric, are also similar to the ones presented in the paper for the baseline case.

[Figure]

Figure 2: Optimum designs for various objective functions and market scenarios (left). MIRR across the entire turbine design space (right) for an extreme market scenario.

Since the focus of both the papers was turbine sizing, the layout optimization was intentionally simplified to avoid additional parameters influencing the result. Also, the layout is only re-adjusted and not optimized, even for LCoE, every time the turbine size changes. The purpose of a regular near-square layout is to avoid any boundary effect that may favor one turbine size over the other. With a standard layout that is always close to a square shape, the influence of turbine size on wake losses is reproducible and mostly monotonic.

As the preliminary results without the power constraint show similar results as the baseline case presented in the paper (w.r.t. the differences between the design), the analysis is not included in the paper. However, if the reviewer finds it necessary to generalize results, this sensitivity study can be added as an appendix.

**Reviewer 2 comment**

In the paper "Designing wind turbines for profitability in the day-ahead markets", the authors present an interesting and timely study of how variable electricity prices and especially negative correlation between prices and wind speed could impact the optimal design of wind turbines. However, I have some questions about the assumptions, and some comments related to the interpretation of the results and the conclusions. I also suggest that some additional case studies/sensitivity studies could help to generalize the findings.

- It is mentioned that the coefficient of variation (CV) is kept constant. What is this constant value? (maybe it's mentioned somewhere, sorry if I missed it. But it would be nice to write it in Table 1 so it's easy to find)

  The constant value used for the analysis is 0.4. It is currently mentioned in the text before Table 1, but we agree that it should be mentioned in Table 1 to make it explicit to the reader.

- The authors say: "It is known that the value for CV also differs, but it is expected to have the smallest range of variability of all the three market parameters". I would slightly disagree. E.g., in the reference Swisher, et al. (2022), CV is expected to roughly double from 2025 to 2045, and similar growth is seen also in other studies (reflecting the increasing share of wind and solar generation with near-zero operating cost, increasing CO2 tax and decommissioning of base load thermal generation, which all tend to drive price variability up). Also, while the mean price can vary a lot between years (e.g., due to weather and gas prices), the increase in CV is usually seen as a more permanent change as we go towards a highly weather dependent system with only a few peaker plants. Thus, I find a bit surprising that CV is not varied (especially to higher values than seen historically). It would be great to hear more justification on this selection or potentially to see a sensitivity study where CV is also varied (especially a high CV and strong negative correlation between wind speed and prices would be an interesting case to study).

  The initial choice of the CV was based on the values currently observed in Denmark, which already has a high wind penetration. The assumption was that most European countries would reach wind penetration levels similar to Denmark, leading to similar CVs Europe-wide. Also, the choice of CV can significantly influence the results for the extreme cases with high mean prices and high levels of correlation. The current choice of 0.4 results in a standard deviation of 40 for a mean spot price of 100. This already results in a high price range of about 0-200 Euros/MWh. At some point, very large price variations might lead to some restoring measures, be it by storage or regulations. Also, if, in the future, CV increases due to other renewables like solar, then the correlation coefficient w.r.t. wind would decrease. However, the point about sensitivity to CV is interesting, and results for a CV of 0.7 are included in the paper.

  5.4 Sensitivity to the coefficient of variation

  With the rise in wind and solar penetration over the next few years, the price variations are expected to increase, as mentioned in Swisher et al. (2022). The effect of the correlation coefficient is amplified for a higher CV value, as shown in Fig. 4 (of the paper). At some point, very large price variations might lead to some restoring measures, be it by storage or regulations. Also, if, in the future, CV increases due to other renewables like solar, then the correlation coefficient w.r.t. wind would decrease. Fig. 3 (to be added as Fig. 16 in the paper) presents results for a CV value of 0.7, higher than the baseline value of 0.4. The optimum designs for all market scenarios and two relevant objectives are shown in Fig. 3 (left). The difference in the absolute value of MIRR between the two designs for a mean spot price of 100 and correlation of -0.5 is shown in Fig. 3 (right).

  As seen in the figures, the differences in the design and the value of the objective are similar to those of the base case. However, the differences are significant for the highly unlikely scenarios of high CV with high levels of anti-correlation. For higher CV values, the standard deviation of the spot prices is also higher, leading to a larger spread in the prices. For high anti-correlation values, this difference in spot prices between lower and higher wind speeds is amplified, further

[Figure]

Figure 3: Optimum designs for various objective functions and market scenarios with a high CV (left). MIRR across the entire turbine design space (right) for an extreme market scenario with a high CV.

favoring low specific power turbines. Hence, redesigning the turbine specifically for the market could be beneficial for future scenarios where the price variations are extremely high, along with high levels of anti-correlation with wind.

This point is also added to the second conclusion.

The benefits of redesigning the turbine for a specific market scenario are marginal. It is seen that even in a market scenario with more than twice as much cannibalization as currently seen anywhere, the values of MIRR and PI for an LCoE-optimized design are 7.5% and 1.86 respectively, while the values for the market-optimized design are 7.6% and 1.89, respectively. The relative differences are insignificant for most market scenarios. However, a market-driven design could potentially be beneficial for future (less likely) scenarios with even more extreme cannibalization.

- The study is performed for a location with high mean wind speed. However, many of the very low specific power turbines in the literature are suggested for lower mean wind speed sites (usually onshore). Do you think that the results could be different for a low/medium wind resource onshore site? (even nicer would be to add such sensitivity study to the paper)

The sensitivity to wind speed is interesting, even from a pure LCoE standpoint. Hence, in the first paper, the sensitivity of the LCoE-optimum to wind speeds was explored. This led the LCoE-optimum itself to be shifted towards low specific power turbines irrespective of the market. The driver there is mainly the AEP produced by the turbine itself and not the value of the kWhs produced. Hence, for a low/medium wind speed site, all the market-driven optimums would further shift towards lower specific power designs along with a shift in the LCoE-optimum. Since this paper focuses on the differences between LCoE-optimized and market-driven designs, the sensitivity to wind speeds is not included. The sensitivity to wind speeds is highly relevant to our previous study, which explicitly focuses on the drivers for turbine design (https://doi.org/10.5194/wes-9-141-2024).

- The paper presents the optimal designs over a design space of rotor diameter and rated power for a "standard turbine". E.g., the LowWind concept presented Swisher, et al. (2022) is somewhat more radical, with the objective of reducing CAPEX by completely shutting down the turbine at a very low cut-off wind speed. Would you see such more dramatic design choices becoming potentially interesting in the future, or would you still say that the "standard turbine" with LCOE-driven optimization is the way to go?

As some preliminary studies have shown, such designs could have potential benefits for the system. To assess the LCoE or profitability of such designs, their costs need to be assessed as well. There are two main propositions for such LowWind concepts to prohibit excessive increase of loads, thus containing their costs. One proposition is to reduce the cut-out wind speed. Loads near cut-out wind speed are only potentially design driving due to dynamic response. Since the models used for this paper are based on static response, the advantage of following this proposition would not be captured. However, the designs evaluated in this paper are therefore also not disadvantaged by their high cut-out wind speed. Thus, since very LowWind concepts do not appear in this study, we do not expect that the reduction in cut-out wind speed will be a crucial enabler for such technology.

Another proposition is a reduction of operational peak loads around rated wind speed. This is the principal idea for the hybrid-lambda rotor (`https://doi.org/10.5194/wes-9-359-2024`). This would certainly affect the design driver and cost driver as implemented in our models. Thus, we do expect that this type of severe peak shaving could lead to cost reductions that would favour larger rotors.

It would be interesting to perform an analysis similar to that shown in this paper with a model for these cost reductions, to assess the magnitude of the effect. However, there would still be design, manufacturing and operational challenges to overcome that would not be captured by our framework. The extent to which such ideas would lead to the lower specific powers that would be desirable for revenues maximisation and system balancing remains speculative to us at the moment. However, the pursuit of peak shaving to enable further increase in rotor diameter would not conflict with the mechanisms that we have identified. It should however be noted that peak shaving leads to some reduction in power performance near rated wind speed, and that this is often around wind speeds with high probability, especially for LowWind concepts.

- A comment which you may consider related to the discussion about "designing for the wrong market". Overall, it's a very important point, and the robustness of the LCOE optimized design is an interesting result. However, while I agree that the future mean electricity price level is pretty much guesswork, I consider the negative correlation between wind speed and prices to be a more consistent expected future, e.g., in the central and northern Europe (as wind generation is expected to dominate and thus negative correlation seems almost inevitable). Thus, the risk of "designing for the wrong market" in the sense of getting the correlation wrong (that it would be around zero rather than, e.g., around -0.5) would seem to me to be a somewhat lower risk than getting the mean price level wrong.

Interesting comment. The absolute value of the metric will depend more on the mean spot price and less on the correlation. The authors agree that we can be more certain about the anti-correlation and less about the mean spot prices. However, for all the profitable scenarios, the correlation shifts the optimum in the direction of changing specific power while the mean spot price shifts in the optimum in the direction of the same specific power. Hence, we would argue that the risk of designing for the wrong correlation is higher than that of designing for the wrong mean.

---

## Referee Report (RR1)

**Referee Report**

I appreciate the effort in making the new simulations (without the power constraint) and the thorough explanation of my initial concerns.

The main question I raised in the first review was how the area constraints affect the optimisation results. The authors argue that most current and planned wind farms in the North Sea have power densities between 5 - 10 MW/m2, and they therefore limit the optimisation to 6.7 MW/m2. I argued that 2.2 MW/m2 might be more realistic for future tender, but I also questioned why a constraint is even needed. The Borssele wind farm that the authors reference has a power density of 4.4 MW/km2. The new North Sea tenders in Denmark have power densities of about 2.6-2.7 MW/km^2 (estimates can be made based on the map material https://ens.dk/sites/ens.dk/files/Vindmoller_hav/nsl_hjemmeside_202402_0.png - 1 GW for each area).

The new simulations added to the review comments show that the constraints are unnecessary and that results are similar to the constrained results. This is a significant result that the authors should (briefly) mention in the Method section to argue that the chosen constraints have minimal effects on the results.

The other question raised in the original review was regarding the gridded placement of the wind turbines, which are aligned to maximise production, not market value. The authors write in the review comments that layouts that maximise production are not expected to favour the LCOE-optimised turbine compared to the market-optimised turbine. I am not convinced about this statement. Wake losses are traditionally measured as a loss in energy production. However, the new perspective of market-optimised designs should see wake losses as a reduction in revenue. Production- and revenue wake losses are not necessarily directionally aligned. The authors should clearly state (or discuss) in the method section that they assume that the production-optimised layout does not favour the LCOE-optimised turbine compared to the market-optimised turbine.

---

## Author Response (AR2)

**Manuscript ID: WES-2024-43**
**Designing wind turbines for profitability in the day-ahead markets**

September 18, 2024

The authors would like to thank the reviewers for examining the revised manuscript and for accepting it as is. Based on the suggestion by Reviewer 1, a couple of sentences have been added to the final version, marked in red.

Added sentences in the methodology section 2.1 under the layout description:
Orienting the diagonal of the layout along the dominant wind direction minimizes the overall wake losses. Since the dominant wind direction is the same for both low and high wind speeds, it is assumed that this layout does not favour the LCOE-optimised turbine compared to the market-optimised turbine

Added paragraph at the end of problem formulation section 3:
It should be noted that both the constraint magnitude and formulation might change in the future. There could be farms with a different power density or without power/area constraints. Mehta et al.(2024) show the sensitivity of the optimum w.r.t. the constraint magnitude and formulation. Although the absolute values of the optimum designs might differ, it is expected that the differences between market-driven designs and the LCoE-optimized design would be less sensitive to constraints.